# Impact of Tranexamic Acid on Chondrocytes and Osteogenically Differentiated Human Mesenchymal Stromal Cells (hMSCs) In Vitro

**DOI:** 10.3390/jcm9123880

**Published:** 2020-11-29

**Authors:** Mike Wagenbrenner, Tizian Heinz, Konstantin Horas, Axel Jakuscheit, Joerg Arnholdt, Susanne Mayer-Wagner, Maximilian Rudert, Boris M. Holzapfel, Manuel Weißenberger

**Affiliations:** 1Department of Orthopaedic Surgery, University of Wuerzburg, Koenig-Ludwig-Haus, Brettreichstr. 11, 97074 Wuerzburg, Germany; m-wagenbrenner.klh@uni-wuerzburg.de (M.W.); t-heinz.klh@uni-wuerzburg.de (T.H.); k-horas.klh@uni-wuerzburg.de (K.H.); a-jakuscheit.klh@uni-wuerzburg.de (A.J.); j-arnholdt.klh@uni-wuerzburg.de (J.A.); m-rudert.klh@uni-wuerzburg.de (M.R.); b-holzapfel.klh@uni-wuerzburg.de (B.M.H.); 2Department of Orthopaedics, Physical Medicine and Rehabilitation, University Hospital, LMU Munich, Marchioninistraße 15, 81377 Munich, Germany; susanne.mayer@med.uni-muenchen.de; 3Regenerative Medicine, Institute of Health and Biomedical Innovation, Queensland University of Technology (QUT), 60 Musk Avenue, Kelvin Grove, Brisbane, QLD 4059, Australia

**Keywords:** tranexamic acid, hMSCs, chondrocytes, osteoarthritis, toxicity, differentiation capacity

## Abstract

The topical application of tranexamic acid (TXA) helps to prevent post-operative blood loss in total joint replacements. Despite these findings, the effects on articular and periarticular tissues remain unclear. Therefore, this in vitro study examined the effects of varying exposure times and concentrations of TXA on proliferation rates, gene expression and differentiation capacity of chondrocytes and human mesenchymal stromal cells (hMSCs), which underwent osteogenic differentiation. Chondrocytes and hMSCs were isolated and multiplied in monolayer cell cultures. Osteogenic differentiation of hMSCs was induced for 21 days using a differentiation medium containing specific growth factors. Cell proliferation was analyzed using ATP assays. Effects of TXA on cell morphology were examined via light microscopy and histological staining, while expression levels of tissue-specific genes were measured using semiquantitative RT-PCR. After treatment with 50 mg/mL of TXA, a decrease in cell proliferation rates was observed. Furthermore, treatment with concentrations of 20 mg/mL of TXA for at least 48 h led to a visible detachment of chondrocytes. TXA treatment with 50 mg/mL for at least 24 h led to a decrease in the expression of specific marker genes in chondrocytes and osteogenically differentiated hMSCs. No significant effects were observed for concentrations beyond 20 mg/mL of TXA combined with exposure times of less than 24 h. This might therefore represent a safe limit for topical application in vivo. Further research regarding in vivo conditions and effects on hMSC functionality are necessary to fully determine the effects of TXA on articular and periarticular tissues.

## 1. Introduction

One of the main goals among orthopedic surgeons remains to decrease the peri-operative risk associated with invasive surgeries while optimizing surgical outcomes and maximizing patient satisfaction. According to the Organization for Economic Co-operation and Development (OECD), hip replacement and knee replacement surgeries (309 and 223, respectively, per 100,000 population in Germany in 2017) are two of the most common orthopedic procedures [1]. A specific area that has received increasing attention in this context is the significant blood loss frequently observed during total hip and knee arthroplasty [2]. Severe peri-operative blood loss is associated with an increased risk for cardiopulmonary complications and prolonged hospitalization, possibly predisposing the patient for nosocomial infections and leading to higher healthcare costs [3,4].

Tranexamic acid (TXA) has emerged as a potential solution to this problem and may significantly decrease blood loss observed during total knee and hip arthroplasty [5,6,7]. TXA is a synthetic derivate of the essential amino acid lysine. It inhibits the conversion from plasminogen to plasmin by blocking the lysine binding site on plasmin and therefore limiting the fibrinolytic effects of plasmin [8]. Systemic intravenous treatment with TXA is widely regarded as safe. The topical intraarticular application is often favored for high-risk patients with cardiovascular or thromboembolic risk factors [6,9].

Regarding topical application of TXA, the question remains whether TXA has toxic effects on articular and periarticular tissues. This is especially important in partial joint surface replacements of the knee and hip joints, which preserve large parts of native hyaline cartilage and joint-forming bone. However, only a limited number of studies has examined the effects of TXA on chondrocytes, providing inconsistent results regarding toxicity and recommendations for clinical use [3,8,10,11,12,13,14]. Interestingly, to date, there are hardly any studies investigating the effects of TXA on mesenchymal stromal cells (MSCs), which can be found in a variety of tissues forming and surrounding the human hip and knee joint [15,16,17]. 

MSCs are multipotent precursor cells with the ability to differentiate into adipocytes, osteoblasts and chondrocytes [18]. Therefore, MSCs are vital for healthy bone homeostasis [19]. Functional progenitor cells and remodeling play major roles in the processes of bone adapting to total hip or knee arthroplasty and are believed to be dysfunctional during post-operative complications such as aseptic loosening of total joint implants [20].

Therefore, the goal of this study was to investigate the possible toxicity of TXA on human chondrocytes and bone marrow-derived hMSCs, which had undergone osteogenic differentiation in vitro. Different concentrations of TXA (no TXA, 10 mg/mL, 20 mg/mL and 50 mg/mL) as well as different exposure times (10 min, 24 h, 48 h) were compared regarding the effects on cell viability, osteogenic differentiation capacity of MSCs and the expression of osteogenic and chondrogenic marker genes. 

## 2. Materials and Methods

### 2.1. Isolation and Culture of Chondrocytes

After informed and written consent and as approved by the University of Wuerzburg’s institutional review board (number of the approval 82/08), femoral bone marrow samples and hyaline cartilage for the isolation of MSCs and chondrocytes were harvested from five patients aged 62 to 79 (mean age 65.8 years) [21,22]. In addition, participating patients agreed to the use of collected and examined surgical waste after undergoing total hip replacement surgery. All patients underwent total hip arthroplasty using the anterior approach, and the hyaline cartilage as well as femoral bone marrow reaming were routinely removed during the surgical procedure. 

Hyaline cartilage was harvested from the femoral head within 12 h using a scalpel. Cartilage samples were then cut into small samples of 1–2 mm^3^ and digested overnight using collagenase (0.175 U/mL; Serva Electrophoresis, Heidelberg, Germany) in Dulbecco’s modified Eagle medium (DMEM)/Ham’s F12 (1:1; Life Technologies GmbH, Thermo Fisher Scientific, Waltham, Massachusetts). The next day the suspended cells were spun, resuspended in standard cell culture medium and seeded in 175 cm^2^ plastic cell culture flasks (Greiner Bio-One GmbH, Frickenhausen, Germany). Chondrocytes were grown in standard culture medium including DMEM/Ham’s F12 supplemented with 10% fetal bovine serum (FBS; Life Technologies GmbH) and 1% penicillin/streptomycin (PS; Life Technologies GmbH). Cells were cultured at a temperature of 37 °C, 5% CO_2_, and medium changes were performed every 3 to 4 days (d) until cells reached confluency. 

After reaching confluency, chondrocytes were detached using trypsin for histological and molecular biological assessments, spun, counted and seeded in six-well plates (Greiner Bio-One GmbH) at a density of 3 x 10^3^ cells per cm^2^. After showing confluent growth in the wells, cells were exposed to different concentrations of TXA (Carinopharm GmbH, Elze, Germany) (no TXA, 10 mg/mL, 20 mg/mL and 50 mg/mL) for 10 min, 24 h or 48 h, while untreated chondrocytes remained as controls. 

### 2.2. Isolation and Culture of Bone Marrow-Derived MSCs 

Bone marrow-derived hMSCs were isolated from the femoral reaming of five patients that underwent total hip arthroplasty. In order to isolate MSCs from bone marrow, tissue samples were washed in DMEM/Ham’s F12. Suspended cells were then spun, resuspended and seeded in 175 cm^2^ plastic cell culture flasks (Greiner Bio-One GmbH). MSCs were grown in standard culture medium until reaching confluency (all Life Technologies, Thermo Fischer Scientific, Dreieich, Germany). 

### 2.3. Osteogenic Differentiation of MSCs

After reaching 70% confluence, MSCs were trypsinated, spun and counted. MSCs for histological and molecular biological assessments were then seeded in six-well plates at a density of 3 x 10^3^ cells per cm^2^. Cultures were cultured at 37 °C, 5% CO_2_, and medium changes were performed every 3 to 4 d. 

After reaching confluency osteogenesis was induced for a duration of 21 d using an osteogenic differentiation medium supplemented with 100 nM dexamethasone, 50 μg/mL ascorbate and 10 mM β-glycerophosphate, as described in our earlier studies [23,24]. Simultaneously controls were grown in standard cell culture medium that lacked the mentioned osteogenic supplements for the same duration of time. 

### 2.4. Tranexamic Acid Treatment of Chondrocytes and MSCs

Cell cultures were exposed to various concentrations of TXA (no TXA, 10 mg/mL, 20 mg/mL and 50 mg/mL). As negative control groups, we maintained undifferentiated hMSCs and chondrocytes that were both not treated with TXA. The length of exposure was either 10 min, 24 h or 48 h. TXA was applied after 21 days of osteogenic differentiation in hMSCs and after cells had reached confluence in chondrocytes. The aim was to investigate possible effects of TXA on cell viability, osteogenic differentiation capacity of bone marrow-derived hMSCs and the expression of osteogenic and chondrogenic marker genes. Osteogenically differentiated hMSCs that were used for Alizarin Red S staining were only exposed to 50 mg/mL TXA. 

The stock solution of TXA (Carinopharm GmbH, Else, Germany) (100 mg/mL) was solved in standard cell culture medium while standard cell culture medium alone served as a negative control. After treatment with TXA, cells were washed in phosphate-buffered saline (PBS) and prepared for further histological, biochemical and molecular biological experiments. Images of chondrocytes were taken straight after the respective treatment period with TXA using an inverted-phase microscope (Carl Zeiss Jena GmbH, Jena, Germany). 

### 2.5. Biochemical Assays

Adenosine 5’-triphosphate (ATP) assays were performed to assess cell proliferation, which correlates with cell viability in monolayer cultures of osteogenically differentiated hMSCs and chondrocytes. Cell viability was measured after 10 min, 24 h and 48 h of treatment with various concentrations (no TXA, 10 mg/mL, 20 mg/mL and 50 mg/mL) of TXA using the CellTiter-Glo^®^ luminescent cell viability assay (Promega, Madison, WI, USA), as previously described [22].

After TXA-treatment, hMSCs and chondrocytes that were used for biochemical investigations were trypsinated and seeded in new 96-well-plates (Greiner Bio-One GmbH) at a density of 3 x 10^3^ cells per cm^2^. 

ATP assays were performed after 10 min, 24 h and 48 h. According to the user’s guide, the cells were mixed with 100 μL of CellTiter-Glo^®^ (Promega GmbH, Mannheim, Germany) reagent, a composition of CellTiter-Glo^®^ substrate with CellTiter-Glo^®^ buffer. Cells were incubated in this reagent for 10 min before luminescence was measured using a plate-reading luminometer (Promega GmbH). 

### 2.6. Alizarin Red S Staining

After 21 d of osteogenic differentiation, Alizarin Red S stainings were performed to determine possible effects of TXA-treatment on the mineralization in hMSC monolayer cultures, as described earlier.^23^ Cells were washed in PBS and fixed in ice-cold methanol for 10 min. Cells were then incubated in 1% Alizarin Red S (1%; Sigma-Aldrich) for 10 min. Following this procedure, cells were washed in PBS 3 times before being air-dried at room temperature. Images of stained cultures were recorded using an inverted-phase microscope. 

Stainings and controls were later quantified using an Alizarin Red S staining quantification assay (ScienCell Research Laboratories, Carlsbad, USA). Cells were collected, placed in acetic acid (10%; Sigma-Aldrich) and collected in microcentrifuge tubes (Sigma-Aldrich). The tubes were then vortexed and heated at 85 °C before being incubated in ice. After this, tubes were centrifuged, and acetic acid was neutralized with ammonium hydroxide (10%; Sigma-Aldrich). Standard solution was created following the ScienCell Alizarin Red S staining quantification assay user’s manual. Then, 150 μL of test solution and 150 μL of standard solution were aliquoted in a 96-well plate. Absorbance at 405 nm was examined with a plate reader (Eppendorf AG, Hamburg, Germany). 

### 2.7. RNA Isolation and Semiquantitative RT-PCR

The gene expression levels of the tissue specific osteogenic marker genes alkaline phosphatase (ALP), collagen type I alpha 1 chain (COL1A1), collagen type X alpha 1 chain (COL10A1) or osteocalcin (OC), as well as the chondrogenic marker genes collagen type II alpha 1 chain (COL2A1), aggrecan (ACAN), sex-determining region Y-box 9 (SOX9) and cartilage oligomeric matrix protein (COMP) were examined using semiquantitative RT-PCR. 

Trizol reagent (Invitrogen) and other purification steps such as DNAse treatment were used to isolate RNA from differentiated hMSCs and chondrocytes following treatment with TXA. All steps were performed as described in the user’s manual of the NucleoSpin^®^ RNA II kit (Macherey-Nagel GmbH & Co. KG, Düren, Germany). For the formation of cDNA, 1 μg of isolated RNA was combined with random hexamer primers (Thermo Fischer Scientific) and Promega^®^ M-MLV reverse transcriptase (Promega GmbH). Following this step, 1 μL of cDNA was used as a pattern of amplification in a 30 μL reaction volume consisting of forward and reverse gene-specific primers (5 pmol each) and GoTaq^®^ DNA polymerase (Promega GmbH). Primer sequences, annealing temperatures and cycle numbers for RT-PCR are listed in Table 1. As pointed out in our previous studies, elongation factor 1α (EEF1A1) was used as the housekeeping gene [24,25].

The final products of RT-PCR were split up using gel electrophoresis on 2% agarose (Biozym Scientific GmbH, Hessisch Oldendorf, Germany) gels including 5 μL per 100 mL GelRed^®^ (Biotium, Fremont, CA, USA). The final products of RT-PCR were then separated through electrophoresis on 2% agarose gels (Biozym Scientific GmbH, Hessisch Oldendorf, Germany) containing 5 μL per 100 mL GelRed^®^ (Biotium, Fremont, CA, USA). The relative expression of chondrogenic and osteogenic marker genes was examined by measuring the band densities for all of the tested genes in comparison to the expression of the housekeeping gene EEF1A1.

### 2.8. Statistical Analysis 

Numeric data from ATP assays, semiquantitative RT-PCR and Alizarin Red S staining quantification assays were expressed as dot plots with each dot representing the value of a single donor sample after treatment with different concentrations of TXA for indicated exposure times. Mean values and standard deviations from ATP assays and semiquantitative RT-PCRs were calculated to determine statistical significance but were not pictured in the dot plots to avoid visual complications. ATP and Alizarin Red S staining quantification assays were performed in triplicate (*n* = 3) and repeated on 5 marrow and cartilage preparations from 5 patients (*n* = 5). RT-PCR was performed in triplicate (*n* = 3) and repeated on 5 marrow and cartilage preparations from 5 patients (*n* = 5). Data were checked for normal distribution using the Kolmogorov–Smirnov and Shapiro–Wilk tests. Statistically significant differences between varying concentrations of TXA (no TXA, 10 mg/mL, 20 mg/mL and 50 mg/mL) and exposure times (10 min, 24 h and 48 h to TXA) were assessed using a multiple paired T-test or the Wilcoxon signed-rank test. *P*-values < 0.05 were considered statistically significant. 

## 3. Results

### 3.1. ATP Assay of Chondrocytes 

Chondrocytes were seeded in monolayer cultures until they reached confluency before being exposed to different concentrations of TXA (no TXA, 10 mg/mL, 20 mg/mL and 50 mg/mL) for 10 min, 24 h and 48 h. We examined the effects of TXA-treatment on cell proliferation rates using the ATP assay (Figure 1), while untreated chondrocytes were maintained as negative control groups (Figure 1, no TXA). 

Dot plots were used to visualize the broad scattering of data, which led to high standard deviations. Independent of the concentration of TXA used, there was a non-significant trend towards lower proliferation rates in chondrocyte cultures when increasing the exposure time to TXA. This decline in cell proliferation was especially visible after treating chondrocytes with 50 mg/mL TXA (Figure 1, TXA 50 mg/mL). 

Similar to the comparison of exposure times, cell proliferation rates were lowest when using higher concentrations of TXA (Figure 1). As a result, long exposure times with high concentrations of TXA led to the strongest decline in cell proliferation. In particular, the mean proliferation rate of chondrocytes was significantly lower after 48 h of treatment with 50 mg/mL (Figure 1, TXA 50 mg/mL) of TXA compared to 48 h of treatment with 20 mg/mL of TXA (Figure 1, TXA 20 mg/mL). 

### 3.2. ATP Assay of Osteogenically Differentiated hMSCs 

After primary hMSCs reached confluent growth, osteogenic differentiation was initiated for 21 d. Following osteogenesis, cell monolayers were exposed to different concentrations of TXA (no TXA, 10 mg/mL, 20 mg/mL and 50 mg/mL) for 10 min, 24 h and 48 h. We examined the effects of TXA-treatment on cell proliferation rates and viability using the ATP assay (Figure 2), while undifferentiated hMSCs served as negative control groups (Figure 2, control). 

Again, dot plots were used to visualize the broad scattering of data, which led to high standard deviations. The proliferation rate was similar in undifferentiated cell groups (Figure 2, undiff.) and cultures that were not treated with TXA (Figure 2, no TXA). Within osteogenically differentiated cultures, proliferation rates declined when extending the exposure time to TXA. The biggest decline in cell proliferation rates was observed between 24 h and 48 h exposure time to TXA, independent of the used concentration. However, the effect of different exposure times of TXA on the proliferation rate of osteogenically differentiated hMSCs did not reach statistical significance when using the same concentration. 

When comparing the effects of different concentrations of TXA on the proliferation rate of osteogenically differentiated hMSCs, proliferation rates were lowest when using 50 mg/mL of TXA independent of the exposure time (Figure 2, TXA 50 mg/mL). However, differences regarding cell proliferation rates when using varying concentrations of TXA were non-significant when limiting the exposure time to 24 h. 

This changed when exposing osteogenically differentiated MSCs to TXA for up to 48 h. Proliferation rates were significantly lower when using 50 mg/mL TXA (Figure 1, 50 mg/mL) compared to 10 mg/mL or 20 mg/mL of TXA (Figure 2, TXA 10 mg/mL, TXA 20 mg/mL). 

### 3.3. Microscopical Examination of Chondrocyte Morphology 

After chondrocytes reached confluence, they were exposed to different concentrations (no TXA, 10 mg/mL, 20 mg/mL and 50 mg/mL) of TXA for 24 h or 48 h. The potential effects of TXA on cell morphology and growth was observed using an inverted-phase microscope (Figure 3). Untreated samples were maintained as negative controls (Figure 3, control).

The pictured control cultures (Figure 3, control) were isolated from three exemplary selected donors and represent chondrocytes at passage one. Independent of TXA treatment, cells showed a compact, fibroblastic and spindle-shaped morphology. There were no visible differences between control cultures and cultures treated with TXA when using 10 mg/mL TXA (Figure 3, 10 mg/mL, 24 h) or 20 mg/mL (Figure 3, 20 mg/mL, 24 h) for 24 h. 

After 48 h of treatment with 20 mg/mL of TXA (Figure 3, 20 mg/mL, 48 h) first qualitative changes in morphology of detached cells could be observed in comparison to control cultures. Similar results were observed in cell cultures after treatment with 50 mg/mL for 24 h (Figure 3, 50 mg/mL, 24 h) or 48 h (Figure 3, 50 mg/mL, 48 h). However, no alteration of cell size was observed in TXA treated cultures. 

### 3.4. Alizarin Red S Staining of Osteogenically Differentiated hMSCs 

After 21 d of osteogenic differentiation, hMSCs were treated with 50 mg/mL TXA for 10 min, 24 h and 48 h. After treatment with TXA, Alizarin Red S stainings were performed to examine osteogenesis in differentiated hMSCs (Figure 4a). Untreated (Figure 4a, no TXA) and undifferentiated negative controls (Figure 4a, control) were maintained.

Undifferentiated control samples showed no positive Alizarin Red S stainings (Figure 4a, undiff.), while all osteogenically differentiated samples showed positive Alizarin Red S stainings after 21 d independent of the exposure time to TXA (Figure 4a, no TXA, 50 mg/mL TXA). The intensity of Alizarin Red S stainings seemed higher in samples derived from patient 2 (Figure 4a, patient 2) and patient 3 (Figure 4a, patient 3). 

Furthermore, no visible changes regarding the Alizarin Red S staining intensity could be observed dependent of the exposure time to 50 mg/mL TXA in osteogenic differentiated hMSCs derived from patient 1 (Figure 4a, patient 1, 50 mg/mL TXA) and patient 3 (Figure 4a, patient 3, 50 mg/mL TXA). When comparing cell cultures treated with TXA to untreated control cultures and undifferentiated cultures, no alteration of cell size was observed. 

The calculation of the Alizarin Red S standard curve revealed a high coefficient of determination R^2^ (Figure 4b). Quantifications of Alizarin Red S stainings showed low concentrations of Alizarin Red S in undifferentiated cultures (Figure 4c, undiff.) in comparison to differentiated cultures independant of treatment with TXA (Figure 4c, no TXA, 24 h TXA). Treatment with 50 mg/mL of TXA for 24 h of TXA did not affect Alizarin Red S concentrations following osteogenic differentiation for 21 d. 

### 3.5. Expression of Chondrogenic Marker Genes in Chondrocytes 

RT-PCR was performed to evaluate changes in the expression of chondrogenic marker genes in chondrocytes after treatment with varying exposure times (10 min, 24 h and 48 h) and concentrations (no TXA, 10 mg/mL, 20 mg/mL and 50 mg/mL) of TXA (Figure 5). The results were shown in dependence of the concentrations (Figure 5a–c) of TXA and exposure times (Figure 5d–f) to TXA. 

There was no clear trend regarding the influence of different concentrations of TXA on the relative expression of the chondrogenic marker genes Col2A1, ACAN, SOX9 and COMP after 10 min of exposure time (Figure 5a). When exposing chondrocytes to TXA for 24 h (Figure 5b) or 48 h (Figure 5c) we observed a non-significant decline of the expression of all chondrogenic marker genes after increasing the concentration of TXA. 

Similar results were visualized when picturing the expression of chondrogenic marker genes in dependence of the exposure time to TXA (Figure 5d–f). The relative expression of examined chondrogenic marker genes decreased when exposing chondrocytes to 20 mg/mL (Figure 5e) or 50 mg/mL of TXA (Figure 5f) for increasing periods of time. In contrast, the treatment of chondrocytes with 10 mg/mL of TXA (Figure 5d) had no effect on the expression of specific marker genes. The mentioned differences regarding the expression of chondrogenic marker genes after treatment with 20 mg/mL (Figure 5e) or 50 mg/mL of TXA (Figure 5f) were not significant.

### 3.6. Expression of Osteogenic Marker Genes in Differentiated hMSCs 

RT-PCR was performed to evaluate changes in the expression of osteogenic marker genes in hMSCs after osteogenic differentiation for 21 d as well as treatment with varying exposure times (10 min, 24 h and 48 h) and concentrations (no TXA, 10 mg/mL, 20 mg/mL and 50 mg/mL) of TXA (Figure 6). The results were pictured in dependence of the used concentrations (Figure 6a–c) of TXA and exposure times (Figure 6d–f) to TXA. 

There was no clear trend regarding the influence of different concentrations of TXA on the relative expression of the osteogenic marker genes ALP, COL1A1, OC and COL10A1 (Figure 6a–c). However, the relative expression of all osteogenic marker genes showed a slight decline for longer exposure times when compared to the groups that were not exposed to TXA (Figure 6b–c).

Furthermore, when exposing osteogenically differentiated hMSCs to 50 mg/dL of TXA, we observed a decline in the expression of all osteogenic marker genes when increasing exposure times (Figure 6f). This decline was not visible when exposing osteogenically differentiated hMSCs to 10 (Figure 6d) or 20 mg/dL of TXA (Figure 6e). None of the mentioned differences regarding the expression of osteogenic marker genes were statistically significant. 

## 4. Discussion

The anti-fibrinolytic effects of TXA have shown promising results for decreasing peri-operative blood loss during knee and hip arthroplasty [5,6,7]. Interestingly, topical application can lower the systemic load and therefore the risk for complications mentioned above while precisely providing high local concentrations of TXA at local sites [26]. Further research also highlights that the topical use of TXA reduces post-operative swelling, inflammation, as well as the risk for hemarthrosis and therefore accelerates the patient’s recovery process [27,28,29,30]. These observations have led to the question whether the use of TXA may also reduce complication rates and improve the recovery process during less invasive surgeries, such as unicompartmental knee arthroplasties or soft tissue surgeries [3,13]. Despite its potential benefits, the possible negative effects of TXA on local cells and tissues such as cartilage, bone or synovial cells are still poorly understood and remain uncertain. 

Effective doses for the topical, intra-articular use of TXA range from 250 mg to 3 g, corresponding to concentrations of TXA between 15 and 100 mg/mL [11,31,32]. When TXA is applied intravenously, doses between 10 and 15 mg/kg body weight lead to plasma concentrations of 18 mg/mL after 1 h that later drop to 5 mg/mL after 5 h [33]. TXA is believed to quickly diffuse into the synovial fluid until concentrations of TXA equal plasma levels [11,28]. Although multiple studies considered the use of lower concentrations of TXA up to 20 mg/mL to be safe, current research does not provide consistent results about the toxic effects of TXA on articular and periarticular tissues [3,8,11,12,13,14,17].

Our present in vitro study shows that both low concentrations of TXA (10 mg/mL) or low exposure times (10 min) to TXA led to no significant effects on cell viability and metabolic activity in chondrocytes and osteogenically differentiated hMSCs. In contrast treatment of both chondrocytes and osteogenically differentiated hMSCs with 50 mg/mL TXA for 48 h led to a significant decrease in cell viability when compared to treatments with 10 mg/mL and 20 mg/mL TXA. However, proliferation rates after treatment with 50 mg/mL TXA were not significantly lower than in untreated groups that served as internal controls. This is most likely due to the small sample sizes. Central to interpreting the data in terms of TXA effects is the assumption that a linear relationship exists between cell number and ATP measurement. A compound which would increase cell size without altering the cytoplasmic concentration of ATP would appear to be less efficacious in an ATP assay [34]. However, no alteration of cell size was observed in TXA treated cultures.

In addition, we did not observe a significant effect of TXA treatment on the expression of chondrogenic and osteogenic marker genes in chondrocytes and osteogenically differentiated hMSCs. Nevertheless, there was a slight but non-significant trend towards a dose and exposure time dependent decrease of the relative expression of osteogenic and chondrogenic marker genes in both cell types after treatment with TXA. Relative gene expression was lowest in cells that were treated with 50 mg/mL of TXA for 48 h. Microscopic images supported our hypothesis that cell viability in chondrocyte monolayers started to decrease after treatment with 20 mg/mL for 48 h and was lowest after treatment with 50 mg/mL TXA. In contrast, treatment with 50 mg/mL of TXA showed no qualitative or quantitative effects on Alizarin Red S stainings of osteogenically differentiated hMSCs independent of exposure times.

In summary, we did not observe clearly significant effects of TXA on cytotoxicity as well as marker gene expression in both cell types. However, our results indicate a slight dose-dependent trend towards a decrease in proliferation rates and marker gene expression in both cell types. Bearing in mind that this trend was non-significant, our findings most likely indicate that concentrations up to 20 mg/mL of TXA may be appropriate for topical use when exposure time is limited to 24 h. 

The hypothesis of a dose and exposure time dependent relationship between TXA and chondrotoxicity was proposed by Parker et al., who described the topical use of concentrations up to 40 mg/mL of TXA as safe with the first negative effects on cell viability being evident at 20 mg/mL at 12 h of exposure [13]. In addition, Ambra et al. and Sitek et al. found no significant correlation between exposure to low concentrations of TXA up to 4 mg/mL for 6 h and decreased chondrocyte viability [12]. Tuttle et al. examined the effects of different concentrations of TXA on bovine cartilage explants as well as chondrocyte monolayers [14]. Similar to our results, concentrations as high as 50 mg/mL had cytotoxic effects on murine chondrocyte monolayer cultures, while lower concentrations such as 25 mg/mL did not affect cell viability significantly [14]. 

Similar to our results, Marmotti et al. reported no clearly significant effects on cytotoxicity as well as no significant effects on the expression of chondrogenic markers and cell morphology after topical treatment of chondrocytes with TXA [10]. Further, McLean et al. noted a decrease in cell viability in chondrocytes, fibroblast-like cells and tenocytes after topical treatment with TXA at concentrations as low as 1 mg/mL for 24 h [3]. 

To date, there very little literature discussing the effects of TXA on differentiated or native hMSCs. However, different studies showed that the cytotoxic effects of TXA may not only affect chondrocytes [3,10]. TXA may partially induce caspase-3-dependant mechanisms and therefore also affect proliferation rates and viability in other cell types [3]. This supports our findings, which showed a decrease in cell viability after osteogenically differentiated hMSCs were exposed to 50 mg/mL of TXA for 48 h. Further, prolonged exposure to TXA for 24 h and 48 h affected the expression of osteogenic marker genes in a dose dependent manner, although these findings were not significant. Possible negative effects on the osteogenic proliferation capacity of hMSCs may influence the risk for aseptic loosening of total joint replacements [20]. 

However, there are limitations to our in vitro study. First and foremost, it must be noted that we only examined a small sample size harvested from female donors. Further we observed a wide scattering of values within the five different donors regarding changes in cell metabolism and marker gene expression, which can be seen in the respective dot plots. Chondrocytes harvested from osteoarthritic joints may show decreased proliferation capacity independent of TXA treatment [35]. Further, a monolayer cell culture model was used. According to Tuttle et al., three dimensional cartilage models may be more resistant to TXA due to the barrier function provided by the three-dimensional extracellular matrix in hyaline cartilage [14]. In addition, TXA’s clearance as well as pharmaceutical kinetics in vivo may be influenced by other tissues than the cells tested in our study, which complicates the interpretation of our in vitro findings [11]. All donor samples were derived from hip joints after total hip arthroplasty. This may limit the transferability of our results towards the use of TXA during soft tissue surgery or unicompartmental knee arthroplasty. 

Although current research, including our study, indicates that concentrations below 20 mg/mL and short exposure times up to 24 h may provide a safe setting for the topical application of TXA, further research regarding more realistic conditions and possible effects on the functional capacity of other periarticular cells, such as hMSCs, are necessary. 

## 5. Conclusions

We did not observe clearly significant effects of treatment with varying concentrations (no TXA, 10 mg/mL, 20 mg/mL and 50 mg/mL) of TXA on proliferation rates and marker gene expression in chondrocytes and osteogenically differentiated hMSCs. Isolated significant effects on cell viability were evident after treatment with 50 mg/mL for at least 48 h, while a first non-significant trend towards decreased cell viability was visible after treatment with 20 mg/mL of TXA. In addition, our results indicated a non-significant trend towards the reduced expression of chondrogenic and osteogenic marker genes after treatment with higher concentrations of TXA for prolonged exposure times. Against this background, our study most likely supports the hypothesis of a safe limit for topical use of TXA marked by concentrations up to 20 mg/mL. Nonetheless, further research, including larger sample sizes and the simulation of in vivo conditions is necessary to fully determine the influence of TXA on the functionality and viability of articular and periarticular tissues. 

## Figures and Tables

**Figure 1 jcm-09-03880-f001:**
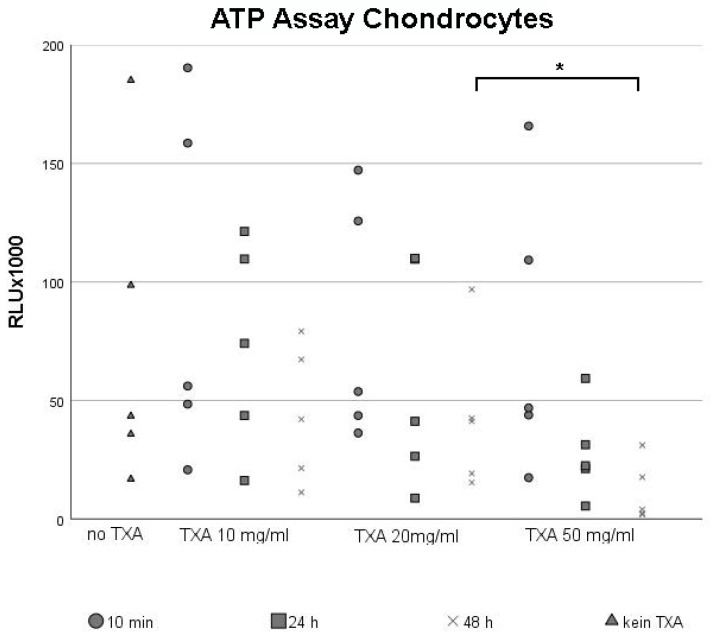
ATP assay of chondrocytes cultured in monolayers after treatment with 0 mg/mL, 10 mg/mL, 20 mg/mL and 50 mg/mL of tranexamic acid for varying exposure times. Chondrocytes from five different donors were seeded in cell culture flasks, incubated in cell culture medium and grown to confluency. After reaching confluency, cells were exposed to 0 mg/mL (no TXA), 10 mg/mL, 20 mg/mL or 50 mg/mL of TXA. The exposure time varied from 10 min to 24 h to 48 h. Chondrocyte proliferation was evaluated using the ATP assay. Each dot represents a single value from all five examined donor samples after treatment with different concentrations of TXA for indicated exposure times. * Significant difference (*p* < 0.05) compared to control samples. TXA, tranexamic acid; min, minutes; d, days; h, hours.

**Figure 2 jcm-09-03880-f002:**
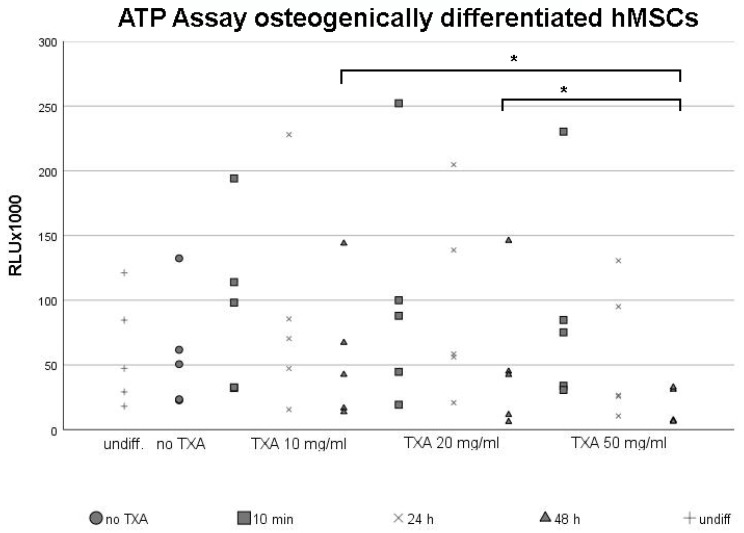
ATP assay of mesenchymal stromal cells following osteogenic differentiation and treatment with 50 mg/mL tranexamic acid for varying exposure times. Osteogenic differentiation was induced in primary hMSCs harvested from five different donors using an osteogenic differentiation medium for 21 d. Undifferentiated control samples (undiff.) were maintained for comparison. After 21 d of osteogenesis, MSCs were exposed to 0 mg/mL (no TXA), 10 mg/mL, 20 mg/mL or 50 mg/m of TXA. Cells were exposed to TXA for 10 min, 24 h or 48 h. Cell proliferation was evaluated using the ATP assay. Each dot represents a single value from all five examined donor samples after treatment with different concentrations of TXA for indicated exposure times. * Significant difference (*p* < 0.05) compared to control samples. hMSCs, human mesenchymal stromal cells; TXA, tranexamic acid; min, minutes; d, days; h, hours.

**Figure 3 jcm-09-03880-f003:**
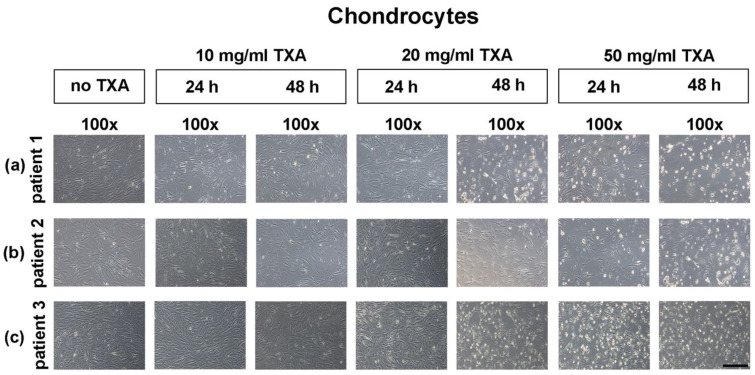
Microscopical analysis of chondrocyte growth in monolayer cultures after treatment with different concentrations of tranexamic acid for varying exposure times. Chondrocytes were harvested from the hyaline hip cartilage of three separate patients that underwent total hip arthroplasty. Cells were seeded in cell culture flasks, incubated in cell culture medium and grown to confluency. After reaching confluency, controls were maintained (no TXA), while other cells were exposed to 10 mg/mL, 20 mg/mL or 50 mg/mL of TXA. The exposure time varied from 24 h to 48 h. Representative samples from three different donors were captured directly after treatment with TXA at low (100×; black bar = 200 μm) magnification. TXA, tranexamic acid; min, minutes; d, days; h, hours.

**Figure 4 jcm-09-03880-f004:**
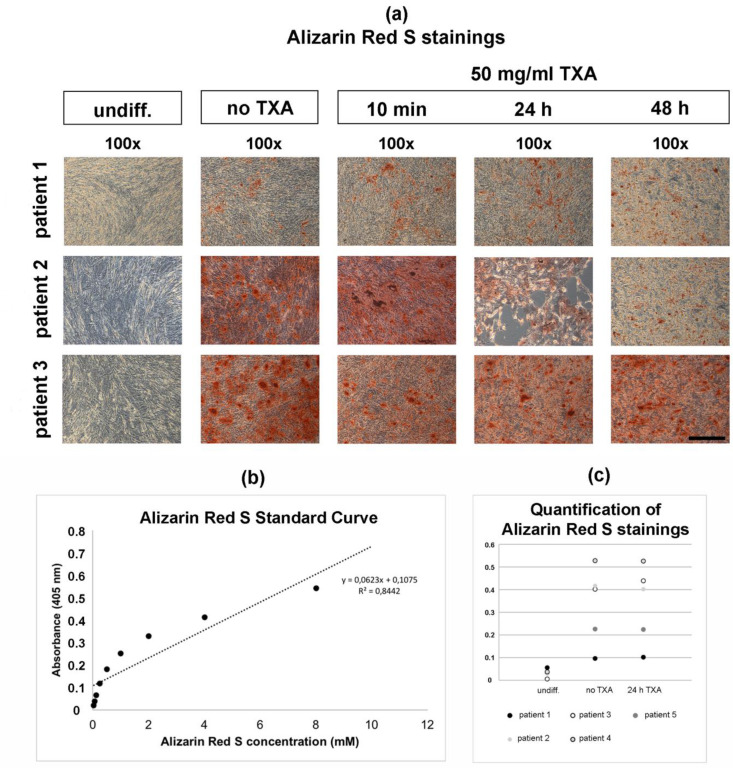
Histological analysis (**a**) of osteogenesis in mesenchymal stromal cells after 21 days in monolayer cell culture and treatment with 50 mg/mL tranexamic acid for varying exposure times. Primary hMSCs were harvested from the bone marrow reaming of three separate patients that underwent total hip arthroplasty. Osteogenic differentiation was induced in hMSCs using an osteogenic differentiation medium for 21 days. Undifferentiated control samples (undiff.) were maintained for comparison. After 21 days of osteogenesis, hMSCs were exposed to no (no TXA), or 50 mg/mL of TXA. Cells were exposed to TXA for 10 min (min), 24 h (24 h) or 48 h (48 h). Alizarin Red S staining was performed to detect extracellular calcium deposits shown as red stainings. Representative samples from three different donors were captured at low (100x; black bar = 200 μm) magnification. The Alizarin Red S standard curve was calculated with the quantification assay of Alizarin Red S stainings (**b**). Quantitative measurements of Alizarin Red S stainings (**c**) were performed in undifferentiated control samples (undiff.), differentiated hMSCs (no TXA), as well as differentiated hMSCs after treatment with 50 mg/mL of TXA for 24 h (24 h TXA). Results were pictured as dot plots. hMSCs, human mesenchymal stromal cells; TXA, tranexamic acid; min, minutes; d, days; h, hours.

**Figure 5 jcm-09-03880-f005:**
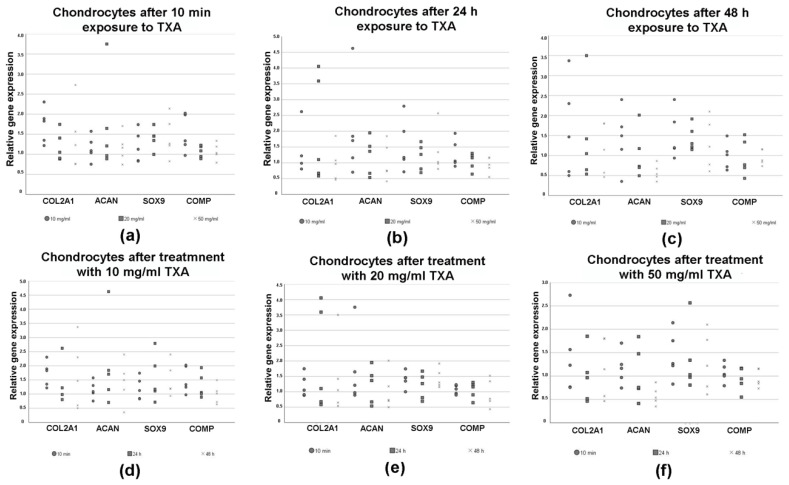
Relative changes in the expression of chondrogenic marker genes pictured as dot plots as measured by semiquantitative RT-PCR in chondrocytes after treatment with 0 mg/mL (no TXA), 10 mg/mL, 20 mg/mL and 50 mg/mL of tranexamic acid for varying exposure times. Chondrocytes were isolated from the hyaline hip cartilage of five patients that underwent total hip arthroplasty. Cells were incubated in cell culture medium and grown to confluency. After reaching confluency, cells were exposed to 0 mg/mL (no TXA), 10 mg/mL, 20 mg/mL or 50 mg/mL of TXA. The exposure time varied from 10 min to 24 h to 48 h. Each dot represents changes in the relative expression of the chondrogenic marker genes aggrecan (ACAN), collagen type II alpha 1 chain (COL2A1), sex-determining region Y-box 9 (SOX9) and cartilage oligomeric matrix protein (COMP) in a single donor sample in dependence of the TXA concentrations (**a**–**c**) and exposure times (**d**–**f**). Eukaryotic elongation factor 1α (EEF1A1) was used as the housekeeping gene and for internal controls. Primer details are illustrated in Table 1. d, days; TXA, tranexamic acid, h, hours; min, minutes.

**Figure 6 jcm-09-03880-f006:**
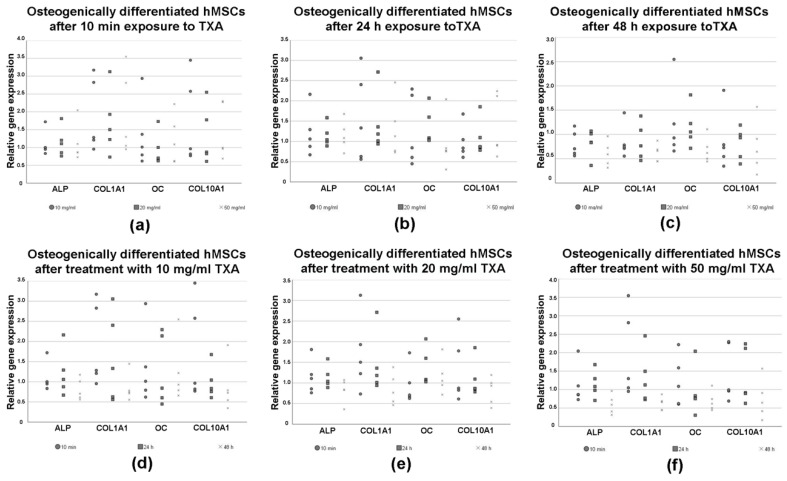
Relative changes in the expression of osteogenic marker genes pictured as dot plots as measured by semiquantitative RT-PCR in osteogenic differentiated mesenchymal stromal cells after treatment with 0 mg/mL, 10 mg/mL, 20 mg/mL and 50 mg/mL of tranexamic acid for varying exposure times. hMSCs were derived from bone marrow reamings of five patients that underwent total hip arthroplasty. Cells were seeded in cell culture flasks and incubated in osteogenic differentiation medium for 21 d. Undifferentiated cells (undiff.) were maintained for comparison. Following osteogenesis cells were exposed to 0 mg/mL (no TXA), 10 mg/mL, 20 mg/mL or 50 mg/mL of TXA. The exposure time varied from 10 min to 24 h to 48 h. Each dot represents changes in the relative expression of osteogenic marker genes collagen type I alpha 1 chain (COL1A1), collagen type X alpha 1 chain (COLXA1), alkaline phosphatase (ALP) and osteocalcin (OC) in a single donor sample in dependance of the TXA concentrations (**a**–**c**) and exposure times (**d**–**f**) are pictured, respectively. Eukaryotic elongation factor 1α (EEF1A1) was used as the housekeeping gene and for internal controls. Primer details are illustrated in Table 1. d, days; hMSCs, human mesenchymal stromal cells; TXA, tranexamic acid, h, hours; min, minutes.

**Table 1 jcm-09-03880-t001:** Primer sequences and product sizes for semiquantitative RT-PCR.

Gene	Primer Sequences (5’–3’)	Annealing Temperature (°C)	Product Size (Base Pairs)	Cycles	MgCl_2_
housekeeping gene for internal control					
EEF1A1	Sense: AGGTGATTATCCTGAACCATCC Antisense: AAAGGTGGATAGTCTGAGAAGC	54.0	234	21	1 x
osteogenic marker genes					
COL1A1	Sense: GGACACAATGGATTGCAAGG Antisense: TAACCACTGCTCCACTCTGG	55.0	461	22	2x
COL10A1	Sense: CCCTTTTTGCTGCTAGTATCC Antisense: CTGTTGTCCAGGTTTTCCTGGCAC	54.0	468	40	1x
ALP	Sense: TGGAGCTTCAGAAGCTCAACACCA Antisense: ATCTCGTTGTCTGAGTACCAGTCC	51.0	454	33	1x
OC	Sense: ATGAGAGCCCTCACACTCCTC Antisense: GCCGTAGAAGCGCCGATAGGC	62.0	293	35	2x
chondrogenic marker genes					
ACAN	Sense: GCCTTGAGCAGTTCACCTTC Antisense: CTCTTCTACGGGGACAGCAG	54.0	400	35	1x
COL2A1	Sense: TTTCCCAGGTCAAGATGGTC Antisense: CTTCAGCACCTGTCCACCA	51.0	155	31	1x
SOX9	Sense: ATCTGAAGAAGGAGAGCGAG Antisense: TCAGAAGTCTCCAGAGCTTG	60.0	263	31	1x
COMP	Sense: CAGGACGACTTTGATGCAGA Antisense: AAGCTGGAGCTGTCTGGTA	54.0	312	32	1x

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
