# Peer review of "Impact of Tranexamic Acid on Chondrocytes and Osteogenically Differentiated Human Mesenchymal Stromal Cells (hMSCs) In Vitro"

_jcm, 2020, doi:10.3390/jcm9123880_

Round 1
Reviewer 1 Report
In the manuscript titled “Impact of tranexamic acid on chondrocytes and osteogenically differentiated human mesenchymal stromal cells (hMSCs) in vitro”, the authors evaluate toxicity of TXA by cell proliferation and cell differentiation. I think that this study has problems as indicated below.
- Table 1 is difficult to see. Please change where the new line starts.
- Does TXA inhibit the conversion from plasminogen to plasmin and limit the fibrinolytic effects of plasmin with the concentration and reaction time used in this study?
Reviewer 2 Report
The authors have investigated the effect of TXA on cultured chondrocytes and BMSC that underwent osteogenic differentiation. Although the setup is sound, the reviewer has multiple methodological remarks that impact the evaluation of the present manuscript.
The type of measurements do not appear to be chosen well, since standard deviations in the figure often exceed the signal of the measurement. As such, the drawn conclusions about the data are valid, but the data quality appears to be quite low. It should be noted that it is not clear whether figures show a n=3 from one donor or multiple donors (5 are mentioned in the methods). In addition, quantification of Alizirin red staining was not done and no follow up of apparent apoptosis was performed, which would be logical to pursue in the view of the reviewer.
Detailed comments:
Page 2, line 53: what intravenous concentrations are achieved with this treatment? And how quickly is it cleared? This would help to assess the choice of dosage in the current in vitro study. 10 mg/ml appears to be very much without this information.
Methods:
What was the solvent of TXA? And what vehicle control was used?
Why were BMSC not expanded in the presence of FGF2? This is generally accepted to facilitate proliferation.
Why did the authors prefer an ATP measurement, which reflects metabolism and cell number over an actual DNA measurement? The latter in the experience of the reviewer leads to very small standard deviations in replicates of the same donor.
Please adhere to HUGO gene nomenclature: which isoform of collagen type I and II was PCRed? The HGNC abbreviation of human Aggrecan is ACAN. Sox-9 should be capitalized to indicate human origin and without the dash. https://www.genenames.org/
Statistical analysis: if experiments were performed on 5 donors, then the statistical power could be increased by using all the data in a paired t-test. It appears that a single donor is shown in the figures with 3 biological replicates? This is also not clarified in the figure legend. A dot plot might facilitate presentation of data from multiple donors.
Results:
Figure 1, 2, 5 and 6 show standard deviations that are almost as high or sometimes even higher as the average measurement. What is graphed in this figure? The average of 5 donors? With how many replicates per donor? If individual donors are graphed, then perhaps a normalized response to control per donor in a dot plot would explain the variation and facilitate interpretation. Currently, the data are nearly uninterpretable, despite the statistical significance for some conditions.
Figure 3: perhaps the 100x in the figure is not so informative.
Since many cells are rounding up in the phase contrast images: why did the authors not try to measure cell apoptosis?
Figure 4: Alizarin red can be easily extracted and quantified in a plate reader. Why was this not done?
Round 2
Reviewer 2 Report
The authors have provided answers to questions of the reviewer and have made some textual clarifications. However, the reviewer still wants to see dot plots of individual donors (averaged) to be able to assess what is going in this dataset. No additional measurements were performed to answer questions of the reviewer and the textual answers to these questions are not satisfactory.
Specific comments:
1)“We chose to picture our data in this manner to combine multiple different informations (TXA concentrations, TXA exposure times, cell types) in one figure. We agree that the use of dot plots would explain the high standard deviations. However, we think that the use of dot plots would probably further complicate the interpretation of data due to the mass of collected informations (TXA concentrations, TXA exposure times, cell types).”
The reviewer does not agree with this assessment. There are only 5 donors, the average of each donor (so 5 dots, instead of 1 bar) should not be an unsurmountable problem for visualization. It would have the added benefit that the reader can assess differences in response between donor (if each dot has a different colour or symbol). It might be possible that one or two donors do not respond and that the majority is convincingly down regulated. This would be the only way to solve the reviewer’s issue with the variation in measurements. Otherwise, I suggest to show these individual plots (in whatever visual representation) to the reviewer to allow proper assessment of the merits of the present study.
2)“High standard deviations were most likely a result of the small study population. Sadly, a bigger study population would most likely significantly increase the financial expenses and scientific efforts necessary for this kind of in vitro study.”
Unfortunately, the presented data does not allow the researchers to draw any firm conclusions. Therefore it is of scientific benefit to improve this aspect of the study. There are many publications where 5 donors give clearly interpretable results.
3) “We strongly agree that the measurement of cell apoptosis would be a great addition for future studies regarding the effects of TXA on cell viability.”
The reviewer was strongly suggesting to perform these additional measurements for the present study. Since the authors indicate that they have this protocol up and running, this should not be a difficult task?
4) “We agree that a quantification of Alizarin Red stainings would have upgraded the quantification of the effects of TXA on osteogenesis in hMSCs. Sadly, we did not have access to the necessary devices in our current study.”
The reviewer is not convinced by this answer, since a simple cuvette or plate reader, capeable of measuring OD at 405 nm and 10% acetic acid, is readily available in most laboratories and is not exactly considered as highly advanced equipment. Is there perhaps a different reason why this is not possible?
